# Different Roles of p62 (SQSTM1) Isoforms in Keratin-Related Protein Aggregation

**DOI:** 10.3390/ijms22126227

**Published:** 2021-06-09

**Authors:** Meghana Somlapura, Benjamin Gottschalk, Pooja Lahiri, Iris Kufferath, Daniela Pabst, Thomas Rülicke, Wolfgang F. Graier, Helmut Denk, Kurt Zatloukal

**Affiliations:** 1Diagnostic and Research Center for Molecular Biomedicine, Institute of Pathology, Medical University of Graz, 8010 Graz, Austria; meghana.somlapura@gmail.com (M.S.); iris.kufferath@medunigraz.at (I.K.); daniela.pabst@medunigraz.at (D.P.); helmut.denk@medunigraz.at (H.D.); 2Gottfried Schatz Research Center for Cell Signaling, Metabolism and Aging, Molecular Biology and Biochemistry, Medical University of Graz, 8010 Graz, Austria; benjamin.gottschalk@medunigraz.at (B.G.); wolfgang.graier@medunigraz.at (W.F.G.); 3School of Medical Science and Technology, Indian Institute of Technology Kharagpur, Kharagpur 721302, India; poojalahiri87@gmail.com; 4Department of Biomedical Sciences, University of Veterinary Medicine Vienna, 1210 Vienna, Austria; thomas.ruelicke@vetmeduni.ac.at

**Keywords:** p62 isoforms, keratins, protein aggregation, protein aggregation diseases

## Abstract

p62/Sequestosome-1 (p62) is a multifunctional adaptor protein and is also a constant component of disease-associated protein aggregates, including Mallory–Denk bodies (MDBs), in steatohepatitis and hepatocellular carcinoma. We investigated the interaction of the two human p62 isoforms, p62-H1 (full-length isoform) and p62-H2 (partly devoid of PB1 domain), with keratins 8 and 18, the major components of MDBs. In human liver, p62-H2 is expressed two-fold higher compared to p62-H1 at the mRNA level and is present in slightly but not significantly higher concentrations at the protein level. Co-transfection studies in CHO-K1 cells, PLC/PRF/5 cells as well as p62^−^ total-knockout and wild-type mouse fibroblasts revealed marked differences in the cytoplasmic distribution and aggregation behavior of the two p62 isoforms. Transfection-induced overexpression of p62-H2 generated large cytoplasmic aggregates in PLC/PRF/5 and CHO-K1 cells that mostly co-localized with transfected keratins resembling MDBs or (transfection without keratins) intracytoplasmic hyaline bodies. In fibroblasts, however, transfected p62-H2 was predominantly diffusely distributed in the cytoplasm. Aggregation of p62-H2 and p62ΔSH2 as well as the interaction with K8 (but not with K18) involves acquisition of cross-β-sheet conformation as revealed by staining with luminescent conjugated oligothiophenes. These results indicate the importance of considering p62 isoforms in protein aggregation disease.

## 1. Introduction

The *SQSTM1* gene encodes the ubiquitously expressed sequestosome-1/p62 (p62), a stress-inducible scaffold protein with a variety of cellular functions. These include autophagy [1], cell signaling in inflammation [2], response to oxidative stress [3], and pathogen infections [4]. Furthermore, p62 has been linked to several human disorders such as cancer [5], Paget’s disease of the bone [6], and protein aggregates occurring in several neurodegenerative diseases as well as alcoholic and non-alcoholic steatohepatitis [7,8].

The first evidence of a role for p62 in protein aggregation diseases was obtained by the identification of p62 as a constant component of intracellular protein aggregates in the liver and neurodegenerative diseases [9,10,11]. For example, hepatocyte-associated p62 aggregates are found as Mallory–Denk bodies (MDBs) in steatohepatitis and as intracytoplasmic hyaline bodies (IHBs) in hepatocellular carcinomas and copper toxicosis, and in the brain as Lewy bodies in Parkinson’s disease, neurofibrillary tangles in Alzheimer’s disease, and huntingtin aggregates in Huntington’s disease [8,12,13,14,15]. These proteinaceous inclusions are rich in amyloid-like structures and are mainly composed of p62, ubiquitin, chaperones, proteasome subunits as well as core proteins characteristic of the inclusion type (e.g., keratin in MDBs, α-synuclein in Lewy bodies or neurofilaments and tau in neurofibrillary tangles) [8,16,17].

p62 consists of several domains involved in protein–protein interactions that may also play a role in protein aggregation, such as the Phox and Bem1p (PB1) domains, which was shown to mediate self-oligomerization of p62, the LC3 interacting region (the LIR responsible for binding to autophagosomes [18]), and the ubiquitin-associated (UBA) domain that binds poly-ubiquitinated proteins [19]. Recently, three different isoforms were identified as being transcribed from the *SQSTM1* gene [20]. The p62 isoform-1 (p62-H1) is the largest protein with 440 amino acids; p62 isoform-2 (p62-H2) differs in the 5’ UTR and uses another translation initiation at methionine 85, resulting in a truncated p62 lacking most parts of the PB1 domain (Figure A1). Furthermore, there is a third transcript variant responsible for the same protein as p62-H2 [21]. Several studies have investigated the role of the PB1 domain, but only few data exist on the expression and functional differences of the different p62 isoforms. One study concentrated on tripartite motif-containing protein 5 alpha (TRIM5α), which interacted with p62-H1 but failed to interact with p62-H2 in HEK 293T and HeLa cells [20]. Regarding the interaction of p62 with LC3, a detailed study with p62-H2 and p62-H1 was performed in HeLa cells, where p62-H2 (unable to oligomerize) was found to interact more strongly with LC3 than p62-H1 and appeared to be present within much larger complexes that consisted of multiple copies of p62 as revealed by FRET analysis [22]. Another study demonstrated that p62 isoforms were proteolytically trimmed by caspase-8 at aspartate 329 in human skin fibroblasts. Since p62-H2 partly lacks the PB1 domain and might not be involved in autophagy, the authors focused on p62-H1; they called the trimmed stable protein p62^TRM^ (1–329aa), which plays a role in mTORC1 activation [23]. 

In order to elucidate the possible functions of p62-H1 and p62-H2 in protein aggregation, particularly regarding hepatocellular inclusions, we investigated their interaction with keratins 8 and 18; in addition, we compared p62-H2 with a SH2 deletion mutant (p62ΔSH2). The SH2 domain is part of the PB1 domain and is also involved in protein–protein interaction. Whereas p62-H1 is ubiquitously expressed in various tissues [24], little is known about the expression pattern of p62-H2 in human tissues [25] and studies have failed to detect a stable p62-H2 protein [21]. We therefore analyzed the expression of p62-H1 and p62-H2 mRNA and proteins in human liver and investigated the aggregation behavior and interaction of p62 isoforms with keratins in different cell types with and without endogenous p62 and keratin, respectively.

## 2. Results

### 2.1. p62-H2 Protein Was Present at a Slightly Higher Level than p62-H1 in Human Liver

In order to quantify mRNA concentrations of p62 isoforms, we designed specific primers for qRT-PCR, where one primer set amplifies only p62 transcript variant 1 (p62 TV1), and the second amplifies all the p62 variants 1, 2, and 3 (p62 TV123) (Appendix A, Figure A1). The purified plasmid DNA of p62-H1 was used to estimate the primer efficiencies for both isoforms (Appendix A, Figure A2). Normalizing p62 isoforms to the house-keeping gene (*β*-actin) and primer efficiencies using MultiD software revealed an approximately three-fold higher concentration of the shorter amplicon derived from all three p62 isoforms than the longer one derived only from p62-H1 mRNA. This indicates that mRNA concentrations of p62 isoforms 2 and 3 (encoding p62-H2) exceed by two-fold the concentration of p62-H1 mRNA in the human liver (Figure 1A). 

In a study with cultured human retinal pigment epithelial (RPE) cells, both p62 mRNA variants were expressed, but no protein corresponding to p62-H2 was detected [24]. In order to investigate the protein expression of p62-H1 and p62-H2 isoforms in human liver, we performed Western blot analyses of three different normal human liver samples using the p62CT antibody, which was raised against the C-terminus of p62 and recognizes both isoforms. Densitometric quantification of protein bands of both isoforms revealed a slightly but not significantly higher expression of p62-H2 compared to p62-H1 (Figure 1B), which is consistent with the mRNA expression. The calculated molecular mass of p62-H1 was 47.7 kDa and for p62-H2 38.6 kDa. We observed a slight difference in the mobility of both isoforms as expected from the calculated molecular masses, which suggests post-translational modifications (Appendix A, Figure A3). The two bands observed might reflect different post-translational modifications of p62-H1 (Figure 1B). To rule out this possibility, we performed Western blot analyses on the same liver samples using a p62 N-terminus (p62NT) antibody that recognizes p62-H1 alone at 62 kDa (Appendix A, Figure A3C). Taken together, our results demonstrate the expression of p62-H1 and p62-H2 isoforms at RNA and protein levels in the human liver.

### 2.2. p62-H2 Formed Larger Aggregates than p62-H1 in CHO-K1 Cells

To investigate aggregation and interaction between overexpressed p62 isoforms and keratins, we performed co-transfection experiments in CHO-K1 cells that lacked an endogenous keratin intermediate filament cytoskeleton. The CHO-K1 cells were previously used to study MDB formation in vitro, particularly the aggregation behavior of p62 and keratins but without consideration of the possible roles of p62 isoforms [26,27]. As a negative control and to demonstrate the possible effects of the stress-induced alterations of endogenous p62 expression, we transfected cells with an empty pCDNA4 vector. This procedure led to the appearance of endogenous p62 as stress bodies (Figure 2A) [1]. 

Overexpression of p62-H1 or p62-H2 alone revealed cytoplasmic inclusions (Figure 2B,C), whereby p62-H2 produced, in repeated experiments, large globular aggregates (Figure 2C) in contrast to the smaller accumulation of p62-H1. The aggregates formed by p62-H2 resembled in their size and shape IHBs found in tumor cells of some HCCs and in hepatocytes in copper toxicosis [8,28]. Interestingly, overexpression of p62ΔSH2 (different from p62-H2 by the presence of the AID domain of PB1) formed only smaller, more granular aggregates (Figure 2D).

Overexpression of keratin 8 (K8) or keratin (K18) alone led to the formation of aggregates that partially co-localized with endogenous p62 stress bodies (Figure 2E,F). In general, K18 aggregates were smaller and more granular than those consisting of K8 (Figure 2E,F). 

Overexpression by co-transfection of p62-H1 together with K8 did not result in co-localization of p62 aggregates with K8 (Figure 2G). In contrast, transfection-induced overexpression of p62–H2 and K8 led to the formation of large aggregates with partial co-localization resembling so-called hybrid inclusions [8], which are regarded as transition stages between IHBs to MDBs (Figure 2H). The p62ΔSH2/K8 co-transfection experiments revealed similar results (Figure 2I) with, however, more granular aggregates than with p62-H2. Similar results were obtained by co-transfection experiments with p62-H2, p62ΔSH2, and K18 (Figure 2K,L). In contrast, p62-H1 and K18 aggregates did not co-localize (Figure 2J). Morphological analyses revealed major differences in the aggregate size among the p62 isoforms. The p62-H2 formed significantly larger and globular aggregates in comparison to the endogenous p62 of the empty-vector-transfected cells and in comparison to p62-H1 and p62ΔSH2, which was consistent in three independent experiments (Figure 2M).

To more precisely assess the extent of interaction and co-localization between the p62-H1 and p62-H2 isoforms with keratins, high-resolution 3D confocal imaging was performed (Figure 3A). Using the Pearson correlation coefficient, we found on a single cell level that transfected K18 was found in close proximity to endogenous p62 aggregates compared to very low interaction with K8 (Figure 3B). While K8 showed a higher degree of co-localization with co-transfected p62-H1 compared with endogenous p62, for K18 a lower proximity to p62-H1 was observed. Nevertheless, both K8 and 18 showed very pronounced interaction with co-transfected p62-H2 (Figure 3C). The results of high-resolution confocal co-localization were further corroborated by wide-field FRET-measurements to increase the spatial resolution of detection of molecular interaction to approximately 20 nm. The FRET measurements revealed results comparable to the Pearson correlation coefficients (Figure 3B,D). Furthermore, the count per cell and morphology of aggregates formed by K8 and K18 as well as p62 isoforms were analyzed. While K8 and K18 formed only a small number of particles in combination with endogenous p62, co-expression with p62-H1 and p62-H2 led to a strong increase in K8 and K18 aggregates. Interestingly, the number of aggregates labeled for p62 in cells co-expressing p62-H1 or p62-H2 did not change significantly compared to endogenous p62 (Appendix A, Figure A3). However, the size of the aggregates strongly increased under conditions of K8/p62-H2 and K18/p62-H2 co-expression. At the same time, K8/p62-H1 and K18/p62-H1 combinations showed either very small size increases, insignificant changes, or even a size reduction compared to K8 and K18 combinations with endogenous p62 (Appendix A, Figure A4). These results indicate a clear-cut involvement of the p62-H2 isoform in facilitating aggregate growth and molecular interaction with keratins, especially K18.

### 2.3. β-Sheet-Conformation Was Predominantly Associated with p62-H2 and K8 Co-Aggregates

Mallory–Denk bodies (containing p62 and keratins as constant components) are prone to acquire cross *β*-sheet conformation [27]. To better understand the mode of interaction of p62 isoforms alone and in the presence of keratins, we used the same set of co-transfection experiments as described in Section 2.2., with additional staining with luminescent conjugated oligothiophenes (LCOs) that generate conformation-sensitive spectral signatures [29]. The p62-H1 alone or in combination with K8 and K18 did not acquire cross-*β*-sheet conformation (Figure 4A). In contrast, p62-H2, p62ΔSH2, and K8 alone (but not K18 alone) acquired cross-*β*-sheet conformation (Figure 4C–F). Interestingly, p62-H2 and some granules of p62ΔSH2 formed structures with cross-*β*-sheet conformation in combination with K8 (Figure 4H,I). The p62-H2 alone as well as in combination with K8 (co-)aggregated, the latter resulting in hybrid inclusions (i.e., the combination of features of IHBs and MDBs) as observed in human liver disease. Although p62-H2 and p62ΔSH2 co-aggregated with K18, the majority of the resulting aggregates did not show cross-*β*-sheet conformation except for a few granular aggregates that were LCO fluorescence positive (Figure 4K,L). It should be stressed in this context that K8, in contrast to K18, is an essential component of MDBs, since the lack of K8 prevents MDB formation [30]. Morphometric analysis demonstrated more ß-pleated sheet conformation in the context of p62-H2 as compared to p62-H1. This effect was even more pronounced in co-aggregates of K8 and p62-H2 (Figure 4M,N).

### 2.4. Overexpressed Ubiquitin Accumulated in Co-Aggregates of Keratins with Both p62 Isoforms

Since misfolded proteins are often ubiquitinated, we performed the same series of experiments, as shown in Figure 4, with additional co-transfection of ubiquitin (triple transfections) in CHO-K1 cells in order to elucidate the role of ubiquitin in the aggregation process of p62 isoforms and keratins. Ubiquitin co-localized with endogenous p62, p62-H1, and p62ΔSH2 (Figure 5A,B,D). Surprisingly, not all aggregates of p62-H2 showed co-localization with ubiquitin: co-localization was observed in smaller aggregates, but weak co-localization occurred in larger globular inclusions (Figure 5C). Interestingly, the majority of the p62-H2 aggregates associated more strongly with the overexpressed ubiquitin compared to p62-H1 and p62 ΔSH2 (Figure 5M). A small percentage of K8 or K18 aggregates showed co-localization with ubiquitin (Figure 5E,F,N). Very weak signs of co-localization of K8/p62-H1 aggregates and ubiquitin were observed (Figure 5G). However, K8 aggregation and co-localization with p62-H2 and p62ΔSH2 were pronounced in the ubiquitin-(co)transfected state (Figure 5H,I,N). Furthermore, K18 aggregates co-localized with p62-H1 in the situation of overexpressed ubiquitin (Figure 5J). This co-aggregation of K18 with p62-H1 was markedly more pronounced than without overexpression of ubiquitin (compare with Figure 3 and Figure 4). The p62-H2 and p62ΔSH2 readily co-localized with K18 regardless of the presence or absence of ubiquitin (Figure 5K,L,N). These results suggest that the role of ubiquitin in the interaction of p62 isoforms with keratins varies depending on the type of aggregate.

### 2.5. p62-H1 and p62-H2 did Not Co-Localize with the Endogenous Keratin Intermediate Filament Cytoskeleton of PLC/PRF/5 Cells

Next, we performed experiments with the human hepatocellular cancer cell line, PLC/PRF/5, that contains an endogenous keratin intermediate filament cytoskeleton. Neither endogenous stress body-like p62 aggregates induced by transfection with empty vector (Figure 6A) nor overexpressed p62-H1, p62-H2, or p62ΔSH2 showed co-localization with the keratin intermediate filament cytoskeleton (Figure 6B–D). The p62-H1 aggregates appeared as multiple, small, and ring-like (hollow) structures (Figure 6B), whereas the p62-H2 and p62ΔSH2 aggregates were larger with homogenous staining (Figure 6C,D). In some cells, they were surrounded by keratin filaments (Figure 6D). Overexpression of K8 resulted in small aggregates associated with keratin filaments without association with endogenous p62 granules (Figure 6E). Overexpression of K18 resulted in some distortion of the keratin filament cytoskeleton and the appearance of endogenous p62 (Figure 6F). 

Double transfection of p62-H1 in combination with K8 or K18 produced ring-like (hollow) aggregates without interference with the endogenous keratin filament cytoskeleton (Figure 6G,J). However, co-transfection of p62-H2 and p62ΔSH2, respectively, with K8 or K18 showed co-localization of keratin with p62 aggregates similar to the observation in CHO-K1 cells (Figure 6H,I,K,L). Large K8 or K18 aggregates were, however, not found in PLC/PRF/5 cells, which might be explained by the fact that, in contrast to CHO-K1 cells, overexpressed K8 and K18 are mostly integrated into the intermediate filament cytoskeleton.

### 2.6. Overexpressed p62-H2 Predominantly Accumulates in a Diffuse Fashion in the Cytoplasm of Fibroblasts

To address whether endogenous p62 interferes with the transfected human p62 isoforms (the relative concentrations of endogenous p62 and transfected p62 can be seen in the Western blots shown in Figure A3), we performed transfection studies in p62 knock-out (p62KO) mouse embryonic fibroblasts (MEFs). To this aim, fibroblasts were isolated from p62 total knockout mice (p62KO) [7] and mice with a floxed p62 gene (p62flox) as controls. The p62flox MEFs transfected with empty pcDNA4 vector developed endogenous p62 stress granules similar to CHO-K1 and PLC/PRF/5 cells (Figure 7A). Transfection with p62-H1 led to the appearance of granular aggregates (Figure 7B), whereas p62-H2-transfected cells showed a predominantly diffuse cytoplasmic distribution in addition to a few small granular aggregates *(*Figure 7C). In contrast, multiple p62 aggregates were observed in the p62ΔSH2-transfected cells (Figure 7D). No co-localization of endogenous p62 with transfected K8 and K18 was observed (Figure 7E,F). Co-transfected p62-H1 co-localized neither with K8 nor with K18 or with a combination of K8/18 (Figure 7G,J,N). The p62-H2 co-transfected with K8 showed a predominantly diffuse distribution of p62 overlapping with K8 aggregates (Figure 7H). Because of the diffuse p62 staining, it was impossible to distinguish overlapping staining from true co-localization. However, in cells co-transfected with p62-H2 and K18, some small aggregates of p62-H2/K18, suggesting co-localization of both proteins, were observed in addition to the diffuse accumulation of p62-H2 (Figure 7K). Co-transfection of p62ΔSH2 with K8 and K18, respectively, showed MDB-like aggregates with co-localization of p62ΔSH2 with K8 as well as K18 (Figure 7I–L). Since neither of the p62 isoform aggregates showed co-localization with regular keratin intermediate filaments in PLC/PRF/5 cells, we tested whether the co-localization of p62 with keratin depended on misfolding of keratin. Therefore, we performed triple-transfections of K8 and K18 together with either p62-H1 or p62-H2; K8 and K18 co-transfection should result in the formation of keratin intermediate filaments instead of the development of aggregates of improperly assembled (misfolded) keratin as is the case if only one keratin protein type is overexpressed. In fact, keratin intermediate filaments generated by overexpression of both K8 and K18 showed neither association with endogenous nor transfected p62 isoforms (Figure 7M–P). However, in the experiment with triple-transfection, K8 + K18 + p62-H2 co-localization cannot be excluded since there was an overlap of signals provided by the densely packed keratin intermediate filaments and the diffusely distributed p62-H2 (Figure 7O).

The same series of experiments were also performed with the p62KO MEFs (Figure 8) to eliminate possible interfering effects of endogenous p62. The absence of p62-containing stress granules in empty pcDNA4 vector-transfected cells reflected the lack of endogenous p62 (Figure 8A). The p62-H1-transfected cells formed small aggregates with different sizes and shapes (Figure 8B), whereas p62-H2-transfected cells showed a diffuse cytoplasmic distribution along with a few small granules (Figure 8C). Transfected p62ΔSH2 formed somewhat larger aggregates than p62-H1 without diffuse cytoplasmic distribution (Figure 8D). Transfection of either K8 or K18 alone revealed multiple small aggregates consisting of improperly folded keratin (Figure 8E,F). Co-transfection of p62-H1 and K8 did not result in co-localization (Figure 8G). However, co-transfection of p62-H1 and K18 led to inclusions (aggregates) with co-localization of both proteins (Figure 8J). Like in p62flox MEFs, transfected p62-H2 was diffusely distributed in the cytoplasm but also showed aggregates in co-localization with transfected K8 (Figure 8H) and K18 (Figure 8K). This was also the case with transfected p62ΔSH2 and K8 (Figure 8I) and K18 (Figure 8L). The experiments with triple-transfection of K8 and K18 (leading to intermediate filaments) and the various p62 expression constructs yielded identical results as obtained with p62flox MEFs (Figure 8M–P).

### 2.7. p62-H2 Predominantly Accumulated in IHBs and Hybrid Inclusions of Human HCC

Cytoplasmic inclusions, such as IHBs, MDBs, and hybrid inclusions (inclusions showing a mixed phenotype of IHB and MDB), may be found in tumor cells of some human HCCs [31]. Intracytoplasmic hyaline bodies mainly consist of p62 and ubiquitin and are devoid of K8 or K18, whereas major components of MDBs are misfolded keratins, p62, and ubiquitin. Since the type of p62 isoform involved in IHBs is yet unclear and to demonstrate whether the aggregates generated in transfected cells in vitro are similar to aggregates in human tissues in vivo, we investigated the presence of p62 isoforms in IHBs. Tissue sections of a typical HCC containing numerous IHBs, hybrid inclusions, and MDBs were immunostained with two different p62 antibodies; the p62CT antibody detects both p62 isoforms, whereas the p62NT antibody detects only p62-H1 (see Figure A3). By comparing the reactivity of these two p62 antibodies, conclusions can be drawn as to which p62 isoform is involved in inclusion body formation. Surprisingly, homogenous staining with p62CT was observed in typical IHBs, but only small granules were detected by p62NT immunostaining, suggesting p62-H2 as the major p62 isoform involved in the pathogenesis of IHBs (Figure 9A,B,G). Intracytoplasmic hyaline bodies do not acquire cross-*β*-sheet conformation as revealed by negative LCO fluorescence signal. In contrast, hybrid inclusions (characterized by positivity for p62CT/p62NT and K8/18) showed a positive LCO signal indicating the presence of cross-*ß*-sheet conformation (Figure 9C,D). Interestingly, MDBs that are constantly positive for keratin are rich in cross-*ß*-sheet conformation and are positive with both p62CT and p62NT antibodies, suggesting a role of p62-H1 and p62-H2 in MDB formation (Figure 9E,F) whereas in IHBs only p62-H2 was present (Figure 9A,B).

## 3. Discussion

p62 is a constant component of different types of inclusion bodies. Its role in their pathogenesis and functional significance is still largely unknown except for some evidence generated in p62 knockout mice [7]. The human p62 isoforms p62-H1 and p62-H2 differ by the absence of a major part of the protein-interacting PB1 domain in p62-H2. This domain is involved in a variety of signaling pathways [32,33,34]. p62-H1 is known to maintain the protein homeostasis in cells [3,35,36]. Its PB1 domain has conserved lysine residues (type I) and OPCA motif (type II), placing it in the type I/II category. It can initiate homo-oligomerization in a front-to-back fashion as well as hetero-oligomerization with other PB1-containing proteins [37]. Due to the defect of the PB1 domain, p62-H2 can be expected to be excluded from, or at least less prone to, protein interactions and related physiologic functions or be responsible for differences in aggregation behavior with certain types of cellular proteins and related physiologic functions.

Our study demonstrates for the first time the expression of the p62 isoform p62-H2 in human liver, and the differences in behavior of isoforms p62-H1 and p62-H2 with regard to aggregation and interaction with keratins in vitro, depending on the cell type, and in human tissue. The aim of our study was to obtain closer insights into mechanistic aspects of inclusion body formation, particularly with respect to the interactions between p62, ubiquitin, and the hepatocellular keratins 8 and 18. In vitro transfection experiments with different cell types were regarded as suitable to achieve this goal and to correlate the findings with normal and pathologic situations in humans.

A general observation was that overexpressed p62-H2 formed larger aggregates than p62-H1 in different cell types and showed a pronounced affinity to improperly assembled and misfolded keratins. In contrast, aggregated p62-H1 did not significantly co-localize with aggregated K8 or K18. The p62-H2 aggregates arising in PLC/PRF/5 cells closely resembled IHBs (i.e., large globular inclusions without keratin content) present in some human HCCs. Furthermore, some transfection experiments performed in CHO-K1 cells reproduced features of hybrid inclusions which are a combination of p62 with aggregated keratins found in some human liver diseases and are regarded as transition stages between IHBs to MDBs [8,9,26,28,38]. Interestingly, there was also a difference in distribution and shape of the aggregates between p62-H2 and p62ΔSH2 in that the aggregates of the latter were similar to MDBs due to the fact of their more granular appearance. The co-aggregation of p62-H2 and p62ΔSH2, respectively, with overexpressed K8 but not with assembled intermediate filaments reflects the MDB pathogenesis in human HCC. K8 in contrast to K18 is an essential component of MDBs, since lack of K8 or excess of K18 prevents MDB formation [30,39]. In the course of MDB formation, K8 (type II keratin) is overexpressed and, thus, the 1:1 relationship of K8 and K18 (type I keratin) partners, which is essential for intermediate filament formation, is disturbed. The reasons for this situation may be overexpression/increased synthesis of K8 or preferred degradation of K18.

Cell type differences in aggregation behavior of overexpressed p62 isoforms were also evident from our fibroblast experiments. Whereas transfection of p62-H1 led to the appearance of granular and ring-like (hollow) aggregates, overexpressed p62-H2 mostly showed diffuse cytoplasmic distribution. Diffuse cytoplasmic p62-specific immunostaining has, although not correlated with isoforms, been observed in human pancreatic *β*-cells [40] and neural cells [41,42]. Its functional significance is yet enigmatic. However, a recent study in SW480 cells demonstrated the role of p62-H1 stress bodies in trafficking RelA to nucleoli under cellular stress conditions, whereas p62-H2, which was diffusely distributed within the cytoplasm, was ineffective in this respect [43]. 

With regard to the different functions and behavior of the two major p62 isoforms, increasing evidence exists that the accumulation of p62 stress bodies has beneficial effects by facilitating the clearance of stress-induced cytotoxic substances. In this context, the PB1 domain and the UBA domain of p62-H1 are required for efficient autophagic disposal of misfolded proteins [44,45]. Moreover, the different binding characteristics of p62-H1 and p62-H2 with aggregated proteins could be due to the fact that cross-*β*-sheet conformation upon aggregation is facilitated by the deletion of the SH2 together with the PB1 domain at the N-terminus of p62 (see Appendix A, Figure A1), as is the case for p62-H2 and p62ΔSH2 [27]. We found that p62-H1-related aggregates did not acquire cross-*β*-sheet conformation (Figure 4A,B,G,J) as revealed by staining with LCO dyes. p62-H2, p62ΔSH2 in combination with K8 (but not with K18) as well as K8 alone (but not K18 alone) were prone to acquire cross-*β*-sheet conformation. Of note, the presence of *β*-sheet structures in p62-H2/K8 aggregates together with transglutaminase-induced cross-linking could be responsible for increased durability and resistance which is also typical for MDBs [7]. Furthermore, cross-*β*-sheet conformation might mediate the binding of other proteins with cross-*β*-sheet conformation to p62-related aggregates [46]. In addition to cross-*β*-sheet conformation, we also found different roles of ubiquitin in aggregate formation, which like p62 is a constant component of cytoplasmic protein aggregate. Interestingly, co-localization of K18 and p62-H1 was markedly enhanced by co-transfection and overexpression of ubiquitin, suggesting that for aggregates involving p62-H1 ubiquitin-mediated protein–protein interaction is a major factor (Figure 5). Whereas for p62-H2, depending on the aggregate type, different combinations of ubiquitin-mediated protein interactions and acquisition of cross *β*-pleated sheet conformation appear to be instrumental as also seen in IHBs (presence of ubiquitin without ß-pleated sheet conformation) and hybrid inclusions (acquisition of ß-pleated sheet conformation).

In conclusion, our results, obtained in a comprehensive series of transfection experiments in comparison with human pathologic situations, demonstrate a decisive role for p62 isoforms in protein aggregation and related diseases.

## 4. Materials and Methods

### 4.1. Cloning

The p62-H1 (NCBI accession no: NM_003900.5, cloned from position 284 to 1355) and p62-H2 (NCBI accession no: NM_001142299.2, cloned from position 271 to 1342) and an empty vector of pEGFP-C1 (Clontech) were digested using *Age1* and *Mfe1* (Fermentas, Vienna, Austria) restriction enzymes, and digested products were separated by gel electrophoresis. The Gel Extraction and DNA Clean-Up Kit (ThermoFisher Scientific, Vienna, Austria) was used to extract and purify DNA fragments according to the manufacturer’s protocol. Purified DNA was ligated using a Rapid Ligation Kit (ThermoFisher Scientific, Vienna, Austria). The cloned p62-H1 and p62-H2 were further sequence verified by Sanger sequencing at Microsynth, Vienna, Austria.

### 4.2. Quantitative Reverse Transcriptase-Polymerase Chain Reaction Analysis (qRT-PCR) of Human Liver Samples

Human liver samples were collected after surgery and immediately snap-frozen. The study was approved by the Ethics Committee of the Medical University of Graz, Austria (EK 20-119 ex 08/09). Total RNA was extracted and purified from cryopreserved tissue using TRIzol reagent (Invitrogen, Vienna, Austria). The RNA concentration was determined using nanodrop spectrophotometry, and the integrity was assessed using a Bioanalyzer. Two micrograms of purified RNA was reverse transcribed into cDNA according to the manufacturer’s protocol (Invitrogen). Subsequently, qRT-PCR was performed on Quant studio 7 Flex (Applied Biosystems, Vienna, Austria) using the Power SYBR Green PCR Master Mix as a detection fluorophore. For better comparison of the expression of p62 splice variants, primer efficiencies were estimated by a standard curve using purified plasmid DNA and normalized using MutliD software. Primer sequences are listed in Appendix A Figure A1. Data are shown as expression ratios of target genes normalized to the expression of *β*-actin as an internal reference in each sample.

### 4.3. Western Blot Analysis

Whole tissue extracts from snap-frozen human liver samples were prepared using radioimmunoprecipitation assay (RIPA) buffer (ThermoFisher Scientific, Vienna, Austria) in the presence of protease and phosphatase inhibitors (Roche, Vienna, Austria). The samples were subjected to the previously described procedure [47]. Detection of blotted protein was performed using the p62CT antibody (Progen, GP62C, 1:1000, Heidelberg, Germany), p62NT antibody (Progen, GP62N, 1:100), and normalized to *β*-tubulin (Cell signaling, 1:1000). The bound antibodies were visualized with a horseradish peroxidase-conjugated secondary antibody (P0141, P0448; DakoCytomation, Dako, Glostrup, Denmark), using an ECL Western Blotting Substrate (Biorad, Hercules, CA, USA) on an ImageQuant LAS 500 gel imaging system (GE Healthcare, Vienna, Austria). Densitometric quantification was performed using Image Lab software (Biorad).

### 4.4. Cell Culture

The PLC/PRF/5 cells were cultured in DMEM containing 10% fetal bovine serum (FBS) and 1% penicillin–streptomycin (Gibco, ThermoFisher Scientific, Vienna, Austria). The CHO-K1 cells were cultured in Ham’s F-12 (Lonza, Vienna, Austria) containing 10% fetal bovine serum (FBS) and 1% penicillin–streptomycin (Gibco). Mouse embryonic fibroblasts (MEFs) were isolated from day 12 mouse embryos with a floxed p62 gene (p62WT) and total p62 knockout mice (p62KO) used in our previous study [7]. The p62WT and p62KO mice were bred and maintained under standard conditions in a mouse house facility. Mouse experiments were approved by the Austrian Federal Ministry of Science, Research, and Economy in compliance with the Austrian Law for Welfare of Laboratory Animals with license number: BMWF-66.010/0114-II/3b/2012 and BMWFp62 66.010/0114-ii/3b/2013. All animals received humane care according to the criteria outlined in the “Guide for the Care and Use of Laboratory Animals” prepared by the National Academy of Sciences, USA, and published by the National Institutes of Health (NIH publication 86–23 revised 1985). The uterus was removed from the pregnant mouse after cervical dislocation. It was briefly disinfected in Wescodyne solution following 70% ethanol and DPBS (Gibco) before dissection in a sterile 10 cm plate. Subsequently, each embryo was cut into small pieces in the presence of 50 μL trypsin and placed in an incubator for 15 min. The MEFs were resuspended in 20 mL of DMEM containing 10% FBS and 1% penicillin–streptomycin, and they were monitored for growth for 3 days and further expanded for transfection experiments. All cells were maintained in the incubator at 37 °C with 5% CO_2_.

#### 4.4.1. Transfection

Four hundred thousand cells were seeded on glass coverslips in Costar 6 well plates (Sigma–Aldrich, Vienna, Austria) 24 h before transfection and allowed to grow to 80% confluency. Cells were transfected at equivalent concentrations with 1.5 μg of DNA to 3 μL of lipofectamine 2000 for 18 h. Cells were transfected with expression constructs containing human Keratin 8 (#18063, Addgene, Watertown, MA, USA), human Keratin 18 (#18064, Addgene), p62H1, p62H2 and p62ΔSH2 mutant deletion of 1–50 bp (NCBI accession no: NM_0039003, cloned from position 85 to 1360), and human ubiquitin (Accession no: M26880, cloned from position 1960 to 2193). All constructs used in the study contained the cytomegalovirus promoter for constitutive expression. Controls were transfected with pcDNA4 using lipofectamine 2000 (Invitrogen, Vienna, Austria). All plasmid constructs used in the study were sequence verified by Sanger sequencing at Microsynth, Vienna, Austria.

#### 4.4.2. Immunocytochemistry of Transfected Cells

Transfected cells grown on glass coverslips were fixed in methanol for 5 min, followed by acetone for 20 s (−20 °C), washed in PBS, and incubated with p62CT antibody (GP62C, Progen, 1:100), K8 and 18 antibodies (#6038 and #61028, Progen, 1:100), and ubiquitin (Z0458, DAKO, 1:100) for 1 h. Coverslips were rinsed with PBS and incubated with a secondary anti-guinea pig antibody (rhodamine red) (#93431, Jackson Immune Research Laboratories, West Grove, PA, USA, 1:200) and anti-mouse Alexa fluor 488 (ThermoFisher scientific, Austria, 1:200) for 30 min at room temperature. Coverslips were washed with PBS and briefly rinsed with 70% ethanol, then air-dried and mounted with mounting medium (#S3023, DAKO). Nikon A1R confocal laser scanning microscopy was used to analyze the images.

#### 4.4.3. Scoring of p62 Aggregate Size

The CHO-K1 cells transfected with empty vector (pcDNA3), p62-H1, p62-H2, and p62∆SH2 were used for scoring the average size of the p62 aggregates in three independent experiments. Images were taken using a Nikon A1R confocal laser scanning microscope at 60× magnification. Fifteen random areas with approximately 150–170 cells were scored per experiment. Images were Otsu auto thresholded, and the area was measured using the particle analyzer implemented in Fiji.

#### 4.4.4. Confocal Co-Localization and Particle Analysis

The CHO-K1 cells transfected with K8 or K18 and p62-H1, p62-H2, or p62∆SH2 were fixed and immunolabeled. High-resolution images of cells were recorded using a confocal spinning disk microscope (Axio Observer.Z1 from Zeiss, Gottingen, Germany) equipped with 100× objective lens (Plan-Fluor ×100/1.45 Oil, Zeiss), a motorized filter wheel (CSUX1FW, Yokogawa Electric Corporation, Tokyo, Japan) on the emission side, an AOTF-based laser merge module for laser line 405, 445, 473, 488, 561, and 561 nm (Visitron Systems), and a Nipkow-based confocal scanning unit (CSU-X1, Yokogawa Electric corporation). AF488 and rhodamine were alternately excited with 488 and 561 nm laser lines, respectively, and emissions were acquired at 353 and 600 nm using a charged CCD camera (CoolSNAP-HQ, Photometrics, Tucson, AZ, USA). The Z-stacks of both channels in 0.2 μm increments were recorded. VisiView acquisition software (Universal Imaging, Visitron Systems) was used to acquire the imaging data. For IHBs, hybrid inclusions and MDB tissues, LSM 510 META (Carl Zeiss, Jena, Germany) confocal laser scanning microscope with 458 nm, 590 nm, and 694 nm filters were used. Images were blind deconvoluted with NIS-elements (Nikon, Austria). The co-localization was determined on a single cell level for cell culture experiments and carried out on the whole image for tissue experiments using ImageJ and the plugin coloc2. The Pearson correlation coefficient was chosen for quantitative comparison. The size of the AF488 and rhodamine-labeled structures were determined using a custom-made ImageJ macro. After using the rolling ball background correction, the Z-stacks were binarized with a combination of global stack comprehending Otsu auto threshold and a local Otsu auto threshold with an *x*/*y*-radius of 10 pixels. The binary Z-stacks were further analyzed using the 3D manager plugin to determine for each cell the count, volume, and surface of AF488 and rhodamine-labeled particles. The particle size of p62 (p62CT and p62NT)-labeled structures in tissue samples was measured with the particle analyzer in ImageJ after background subtraction using the rolling ball method and Otsu auto thresholding for binarization.

#### 4.4.5. FRET Microscopy and Analysis

The FRET imaging was performed on an inverted wide-field microscope (Observer.A1, Carl Zeiss GmbH, Vienna, Austria) equipped with a 40× objective (Plan Apochromat 1,3 NA Oil DIC (UV) VIS-IR, Carl Zeiss GmbH) and a standard GFP/RFP filtercube (HC 493/574 (GFP/DsR)). Illumination of AF488 and rhodamine was performed at 470 or 561 nm excitation using a pE-2 LED-illumination system (CoolLED Ltd., Andover, UK), and emissions were collected with a beam splitter (T565lpxr) on two sides of the camera. For simultaneous measurements, AF488 and rhodamine were excited for 500 ms each at 470 and 561 nm and images were recorded with a charged-coupled device (CCD) camera (Coolsnap Dyno, Photometrics, Tucson, AZ, USA). Data acquisition and control of the fluorescence microscope setup was performed using the NIS-Elements AR software (Nikon, Vienna, Austria). A custom-made semi-automated ImageJ-macro was used to select each cell and measure the AF488 and rhodamine signals for long and short-pass fluorescence signals. The FRET signals were corrected for bleed-through and crosstalk using single-labeled AF488 and rhodamine-stained samples. To determine the static FRET, the AF488 to rhodamine FRET signals were normalized to acceptor and donor fluorescence.

#### 4.4.6. Triple Co-Localization of K8/18 and p62 with LCO/Ubiquitin 

The LSM 510 META (Carl Zeiss, Jena, Germany) confocal laser scanning microscope equipped with a 32-element photomultiplier tube array detector and 458/488 nm, 590 nm, and 694 nm filters was used to acquire triple-staining images. Confocal images of K8/18, p62 (p62CT and p62NT for tissues), and LCO or ubiquitin-labeled cells were analyzed. First, the K8/18, p62, and LCO or ubiquitin channels were Otsu auto-thresholded. Afterwards, the thresholded K8/18 and p62 channels were combined by multiplying both channels to gain information about the regions of K8/18 and p62 co-localization. Next, K8/18 and p62 co-localization regions were investigated regarding the co-localization with LCO or ubiquitin using the coloc2 tool in ImageJ. The resulting Manders coefficient was used to quantify the triple co-localization of K8/18, p62, and LCO or ubiquitin. 

#### 4.4.7. Immunostaining of Tissues

Human HCC tissue cryosections (4 µm) were fixed with acetone (10 min), air-dried, and stained with LCO dyes (provided by P. Nilsson) for 20 min, followed by staining with antibodies against p62CT, p62NT, and K8/18 (1:100, 1 h incubation). Rinsed with PBS and incubated with secondary antibodies anti-guinea pig antibody (rhodamine red), anti-mouse Alexa fluor 488, and anti-mouse Alexa fluor 647 (1:200, 30 min incubation). Additionally, tissue sections were incubated with DAPI for 5 min prior to mounting. 

#### 4.4.8. Statistical Analysis

Graphs and statistical analyses were performed using GraphPad Prism 7 (San Diego, CA, USA). Student’s *t*-tests were performed on qRT-PCR and Western blot data; *p* < 0.05 was considered significant. One-way ANOVA was performed for scoring of p62 aggregate size, *p*-values (* *p* ≤ 0.05). Kolmogorov–Smirnov tests were performed for co-localization, particle size, and FRET analysis *p*-values (* *p* ≤ 0.05).

## 5. Conclusions

This is the first demonstration of the expression of different p62 isoforms in human liver and their functional relevance in protein aggregation and interaction with keratin both in vitro and in human liver tissues.

p62-H1 and p62-H2 isoforms have different properties in protein aggregation;p62-H2 forms larger aggregates in CHO-K1 and PLC/PRF/5 cells and is present in IHBs in human liver diseases;p62-H1 forms smaller aggregates in vitro and is present together with p62-H2 in MDBs in human liver disease;p62-H2 and p62-H1 differ in their ability to acquire cross-*β*-sheet conformation in protein aggregates.

Consequently, the role of different p62 isoforms should be considered in future studies concerning p62 in protein homeostasis and protein aggregation diseases.

## Figures and Tables

**Figure 1 ijms-22-06227-f001:**
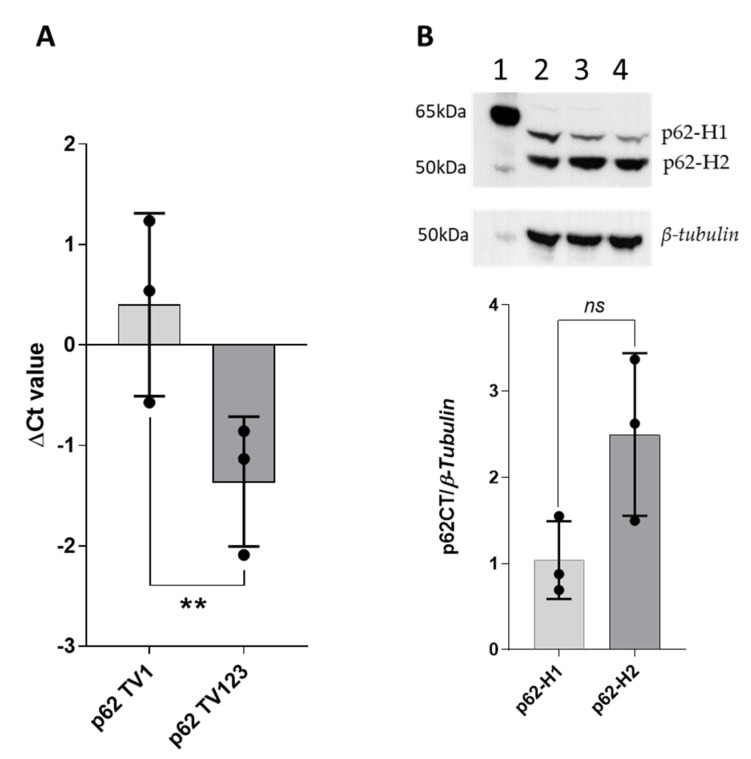
Predominantly expressed p62 isoforms in the human liver. (**A**) mRNA expression of p62 isoform-1 (p62TV1) and p62 isoform-1, 2, and 3 (p62TV123) normalized to *β-actin.* Inter-plate calibration and normalization of primer efficiencies using MultiD software. (**B**) Western blot and densitometric quantification of p62 isoform proteins p62H1 and p62-H2 in three independent human liver samples (40 µg/lane) normalized to *β-tubulin* (Lane 1: protein ladder, 2–4: human liver samples). Immunoblotting using the p62CT antibody that recognizes both isoforms. A paired student’s *t*-test was performed to evaluate significance, ns *p* > 0.05, ** *p* < 0.05.

**Figure 2 ijms-22-06227-f002:**
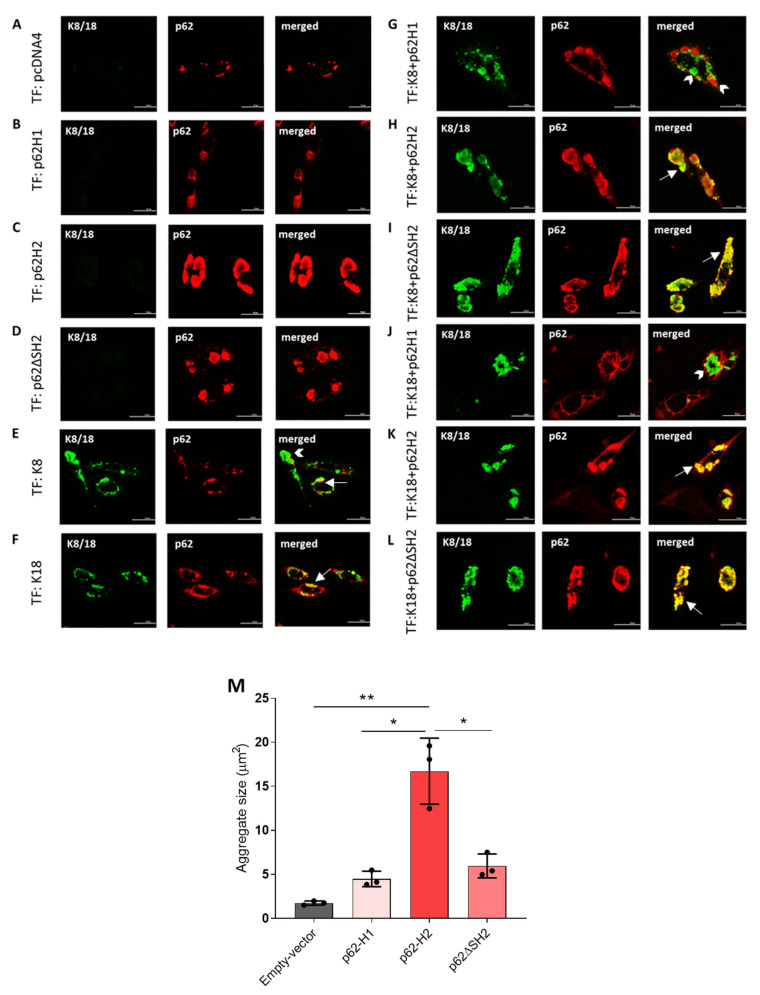
The p62-H2 formed large aggregates and co-localized with K18 in the CHO-K1 cells. The CHO-K1 cells were transiently (co-)transfected (TF) with (**A**) empty vector (pcDNA4), (**B**) p62 isoform1 (p62-H1), (**C**) p62 isoform2 (p62-H2), (**D**) p62ΔSH2 mutant, (**E**) keratin 8 (K8), (**F**) keratin 18 (K18), (**G**) p62-H1 and K8, (**H**) p62-H2 and K8, (**I**) p62ΔSH2 and K8, (**J**) p62-H1 and K18, (**K**) p62-H2 and K18, and (**L**) p62ΔSH2 and K18. Transfected cells were stained for double-label immunofluorescence microscopy with antibodies against p62 (red) and K8/18 (green). Arrows indicate examples of aggregates showing co-localization of p62 and keratin; arrowheads indicate no co-localization. Scale bar = 20 μm. (**M**) The average size of the aggregates in the 130–170 cells/experiment, derived from three independent transfections (*n* = 3), was analyzed using ImageJ software, and the individual aggregate size was assessed. Accordingly, significant differences were assessed by comparing the average aggregate size of p62-H1, p62-H2, and p62ΔSH2 and endogenous p62 (i.e., stress bodies; transfected with empty vector) using one-way ANOVA and Tukey’s multiple comparisons and presented as specific *p*-values (* *p* ≤ 0.05, ** *p* ≤ 0.01).

**Figure 3 ijms-22-06227-f003:**
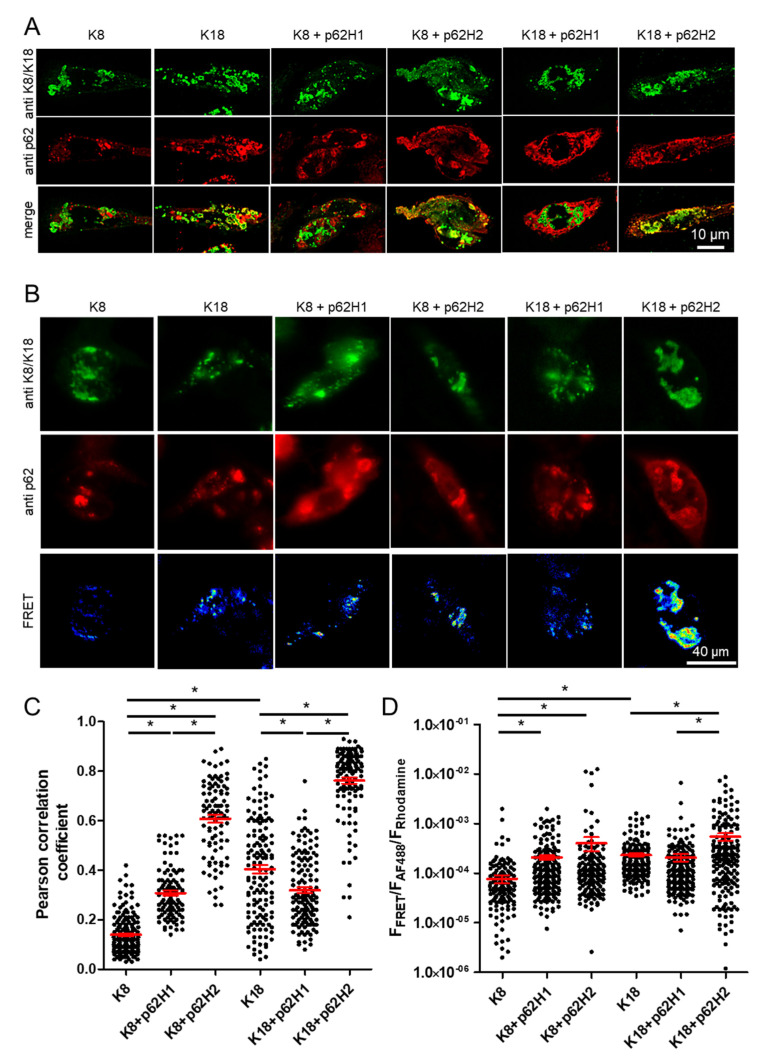
Pronounced molecular interaction of keratins with p62-H2. The CHO-K1 cells were transiently co-transfected (TF) with K8, K18, p62-H1 and K8, p62-H2 and K8, p62-H1 and K18, and p62-H2 and K18 and were stained for double-label immunofluorescence microscopy with antibodies against p62 (red) and K8/18 (green). (**A**) Representative slides of confocal Z-stacks are shown for each condition. (**C**) Co-localization analysis of confocal Z-stacks using the Pearson correlation coefficient of the specimens is shown in a large number of cells (K8, *n* = 65; K8 + p62H1, *n* = 100; K8 + p62H2, *n* = 93; K18, *n* = 139; K18 + p62H1, *n* = 124; K18 + p62H2, *n* = 128). (**B**) Specimens shown in (**A**) were imaged using a widefield FRET system. Representative images for all conditions are shown for p62-labeled (red) and K8/18 (green), and additional FRET acquired signals are shown in pseudo-color. (**D**) Close molecular interaction between K8 and K18, respectively, with endogenous or transiently co-transfected p62-H1 or p62-H2 (K8, *n* = 169; K8 + p62H1, *n* = 187; K8 + p62H2, *n* = 160; K18, *n* = 199; K18 + p62H1, *n* = 202; K18 + p62H2, *n* = 193) using a FRET setup. Single-cell data are shown as scatterplots with mean +/–SEM in red. The results originate from three independent preparations performed on three different days. Using the Kolmogorov–Smirnov test for normality, the results showed no normal distribution. Significant differences were assessed by repeated Kruskal–Wallis tests and Dunn’s multiple comparison tests and presented as specific *p*-values (* *p* ≤ 0.05).

**Figure 4 ijms-22-06227-f004:**
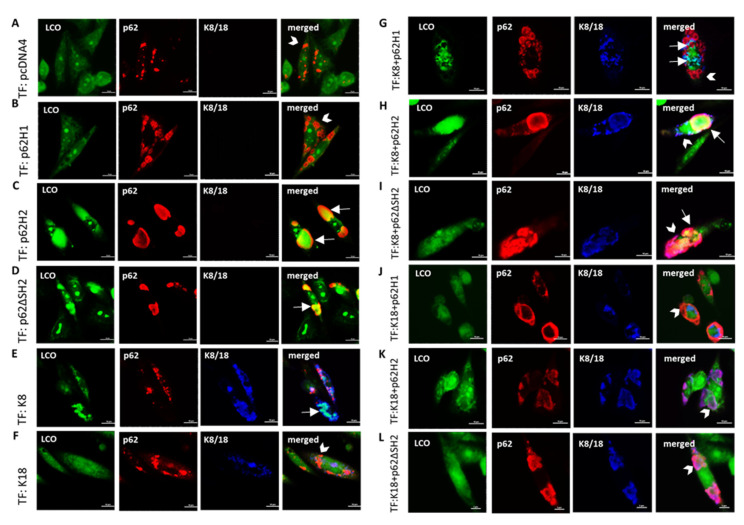
Cross**-***β*-sheet conformation in aggregates demonstrated by using LCO. The CHO-K1 cells were transiently co-transfected (TF) with (**A**) empty vector (pcDNA4), (**B**) p62 isoform1 (p62-H1), (**C**) p62 isoform2 (p62-H2), (**D**) p62ΔSH2 mutant, (**E**) keratin 8 (K8), (**F**) keratin 18 (K18), (**G**) p62-H1 and K8, (**H**) p62-H2 and K8, (**I**) p62ΔSH2 and K8, (**J**) p62-H1 and K18, (**K**) p62-H2 and K18, and (**L**) p62ΔSH2 and K18, and all were stained for triple-label immunofluorescence microscopy with antibodies against p62 (red), K8/18 (magenta), and LCO dye (green). Arrows indicate examples of aggregates showing co-localization of p62 and keratin with an LCO-fluorescence signal; arrowheads indicate examples of no co-localization. Scale bar = 20 μm. (**M**) Co-localization of p62-H1, p62-H2 or p62∆SH2 with LCO-fluorescence in protein aggregates of two independent experiments shown in (B–D) (p62-H1, *n* = 34; p62-H2, *n* = 41; p62ΔSH2, *n* = 41). (**N**) Triple co-localization analysis of K8/18 and p62 overlaying clusters with LCO fluorescence of the specimens shown in (**E**–**L**) (K8, *n* = 30; K8 + p62-H1, *n* = 25; K8 + p62-H2, *n* = 30; K8 + p62ΔSH2, *n* = 32; K18, *n* = 35; K18 + p62-H1, *n* = 32; K18 + p62-H2, *n* = 45; K18 + p62ΔSH2, *n* = 38). Single-cell data are shown as boxplots. Horizontal lines represent the median, the lower and upper hinges show the first quartile and third quartile, and the lower and upper whiskers encompass 10% and 90% of values. Outliers are marked as dots. Using the Kolmogorov–Smirnov test for normality, the results showed no normal distribution. Significant differences were assessed by repeated Kruskal–Wallis tests and Dunn’s multiple comparison tests and presented as specific *p*-values (* *p* ≤ 0.05).

**Figure 5 ijms-22-06227-f005:**
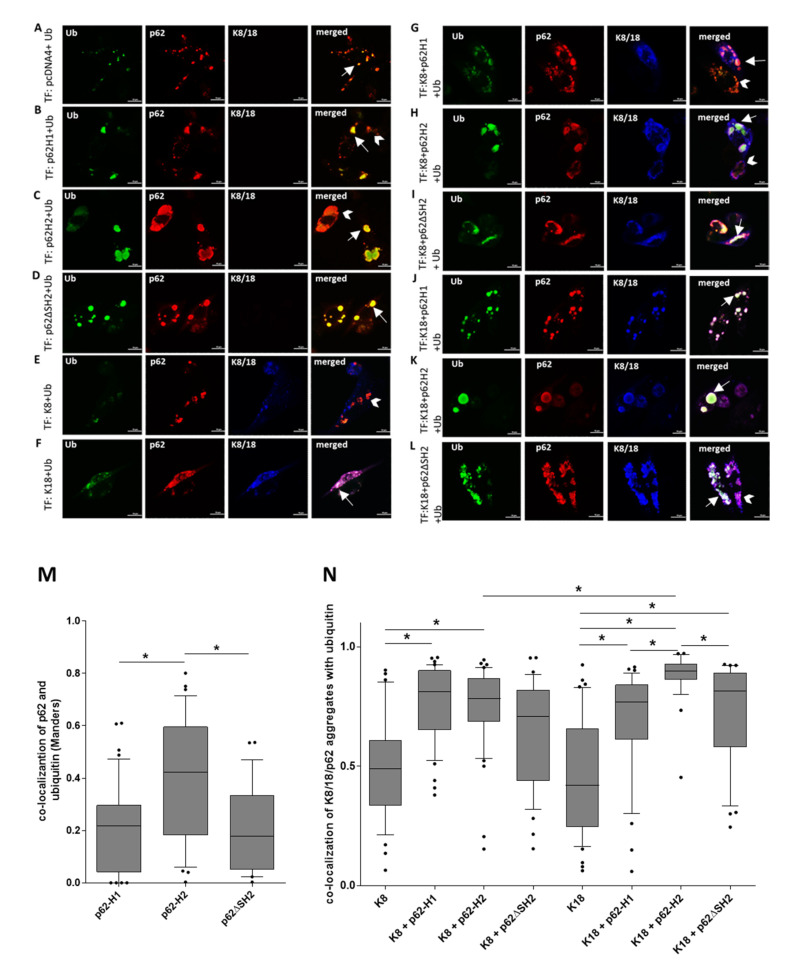
Keratin-8 and keratin-18 co-aggregation with p62-H1 was enhanced by ubiquitin. The CHO-K1 cells were transiently (co-)transfected (TF) with ubiquitin in combination with (**A**) empty vector (pcDNA4), (**B**) p62 isoform1 (p62-H1), (**C**) p62 isoform2 (p62-H2), (**D**) p62ΔSH2 mutant, (**E**) keratin 8 (K8), (**F**) keratin 18 (K18), (**G**) p62-H1 and K8, (**H**) p62-H2 and K8, (**I**) p62ΔSH2 and K8, (**J**) p62-H1 and K18, (**K**) p62-H2 and K18, and (**L**) p62ΔSH2 and K18, and all were stained for triple-label immunofluorescence microscopy with antibodies against ubiquitin (green), p62 (red), and K8/18 (magenta). Arrows indicate examples of aggregates showing co-localization of p62 and keratin with ubiquitin; arrowheads indicate examples of no co-localization. Scale bar = 20 μm. (**M**) Co-localization of p62-H1, p62-H2, or p62∆SH2 with ubiquitin in protein aggregates of two independent experiments shown in (**B**–**D**) (p62-H1, *n* = 41; p62-H2, *n* = 34; p62ΔSH2, *n* = 25). (**N**) Triple co-localization analysis of K8/18 and p62 overlaying clusters with ubiquitin staining is shown in (E-L) (K8, n = 33; K8 + p62-H1, *n* = 42; K8 + p62-H2, *n* = 40; K8 + p62ΔSH2, *n* = 37; K18, *n* = 46; K18 + p62-H1, *n* = 39; K18 + p62-H2, *n* = 27; K18 + p62ΔSH2, *n* = 32). Single cell data are shown as boxplots. Horizontal lines represent the median, the lower and upper hinges show, respectively, the first quartile and the third quartile, and the lower and upper whiskers encompass 10% and 90% of values. Outliers are marked as dots. Using the Kolmogorov–Smirnov test for normality, the results showed no normal distribution. Significant differences were assessed by repeated Kruskal–Wallis tests and Dunn’s multiple comparison tests and presented as specific *p*-values (* *p* ≤ 0.05).

**Figure 6 ijms-22-06227-f006:**
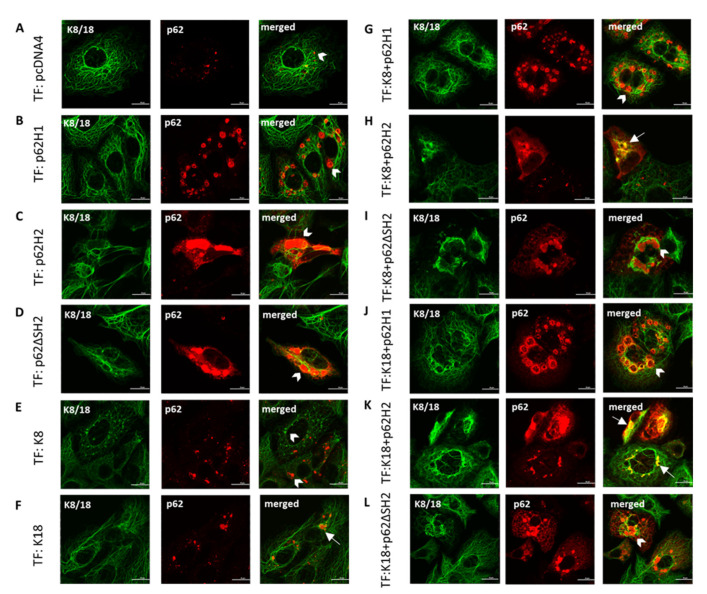
The p62-H2 formed large aggregates and interacted with aggregates of overexpressed K8 and K18 but not with intermediate filaments in PLC/PRF/5 cells. The PLC/PRF/5 cells were transiently (co-)transfected (TF) with (**A**) empty vector (pcDNA4), (**B**) p62 isoform1 (p62-H1), (**C**) p62 isoform2 (p62-H2), (**D**) p62ΔSH2 mutant, (**E**) keratin 8 (K8), (**F**) keratin 18 (K18), (**G**) p62-H1 and K8, (**H**) p62-H2 and K8, (**I**) p62ΔSH2 and K8, (**J**) p62-H1 and K18 (**K**), p62-H2 and K18, and (**L**) p62ΔSH2 and K18, and all were stained for double-label immunofluorescence microscopy with antibodies against p62 (red) and K8/18 (green). Two independent experimental series were performed. Arrows indicate examples of aggregates showing co-localization of p62 and keratin; arrowheads indicate examples of no co-localization. Scale bar = 20 μm.

**Figure 7 ijms-22-06227-f007:**
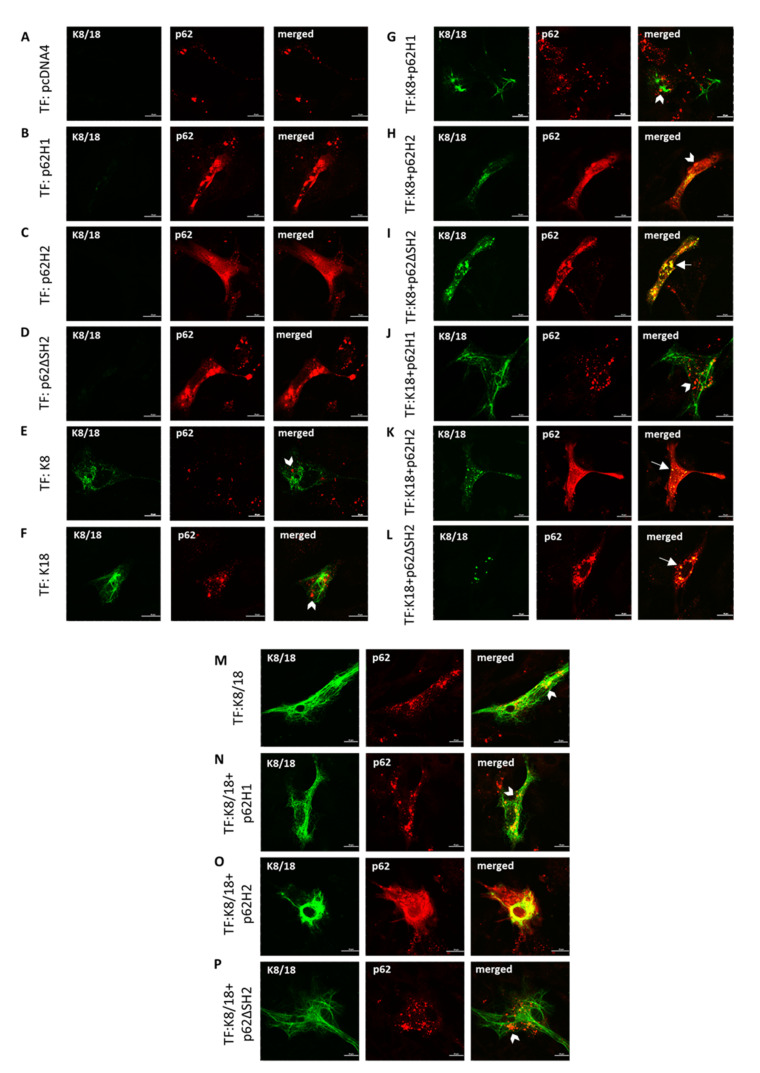
The p62-H2 was diffusely distributed and formed aggregates only in the presence of either K8 or K18 but not of keratin intermediate filaments in p62flox MEFs. Isolated p62flox fibroblasts were transiently (co-)transfected with (**A**) empty vector (pcDNA4), (**B**) p62 isoform1 (p62-H1), (**C**) p62 isoform2 (p62-H2), (**D**) p62ΔSH2 mutant, (**E**) keratin 8 (K8), (**F**) keratin 18 (K18), (**G**) p62-H1 and K8, (**H**) p62-H2 and K8, (**I**) p62ΔSH2 and K8, (**J**) p62-H1 and K18, (**K**) p62-H2 and K18, (**L**) p62ΔSH2 and K18, (**M**) K8 and K18, (**N**) p62-H1, K8 and K18, (**O**) p62-H2, K8, and K18, and (**P**) p62ΔSH2, K8 and K18, and all were stained for double-label immunofluorescence microscopy with antibodies against p62 (red) and K8/18 (green). Two independent experimental series were performed. Arrows indicate examples of aggregates showing co-localization of p62 and keratin; arrowheads indicate examples of no co-localization. Scale bar = 20 μm.

**Figure 8 ijms-22-06227-f008:**
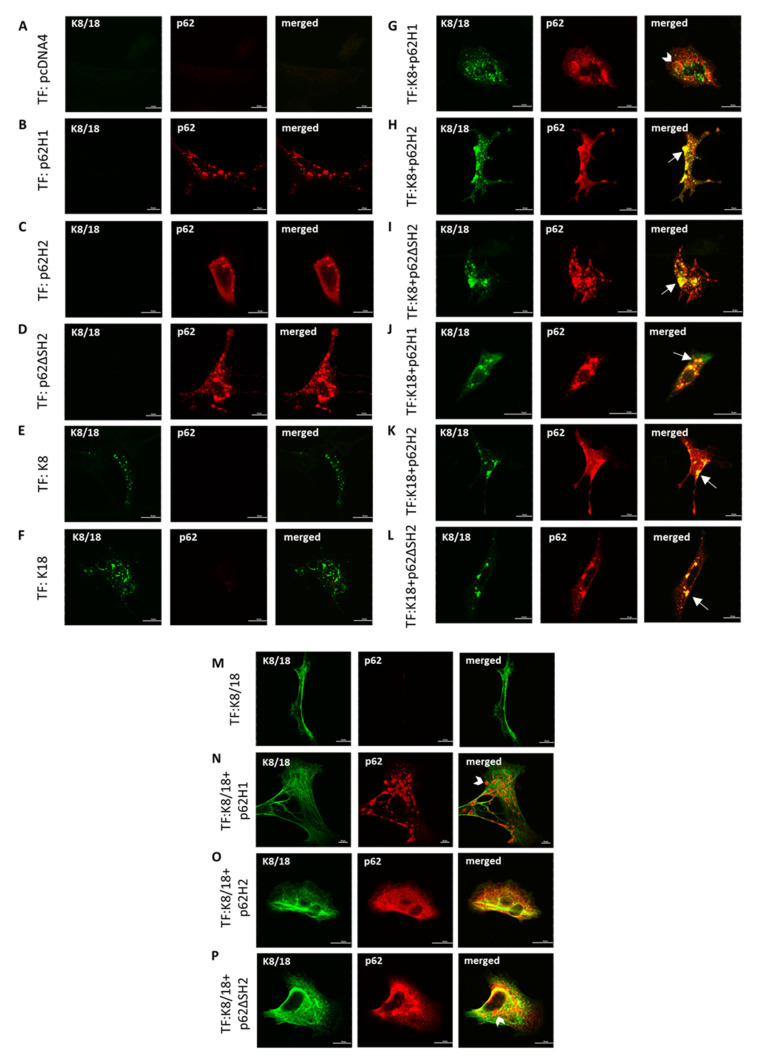
Interaction of p62-H2 with keratin was independent of endogenous p62 in p62KO MEFs. Isolated p62KO MEFs were transiently (co-)transfected with (**A**) empty vector (pcDNA4), (**B**) p62 isoform1 (p62-H1), (**C**) p62 isoform2 (p62H2), (**D**) p62ΔSH2 mutant, (**E**) keratin 8 (K8), (**F**) keratin 18 (K18), (**G**) p62-H1 and K8, (**H**) p62-H2 and K8, (**I**) p62ΔSH2 and K8, (**J**) p62-H1 and K18, (**K**) p62-H2 and K18, (**L**) p62ΔSH2 and K18, (**M**) K8 and K18, (**N**) p62-H1, K8, and K18, (**O**) p62-H2, K8, and K18, and (**P**) p62ΔSH2, K8, and K18, and all were stained for double-label immunofluorescence microscopy with antibodies against p62 (red) and K8/18 (green). Two independent experimental series were performed. Arrows indicate examples of aggregates showing co-localization of p62 and keratin; arrowheads indicate examples of no co-localization. Scale bar = 20 μm.

**Figure 9 ijms-22-06227-f009:**
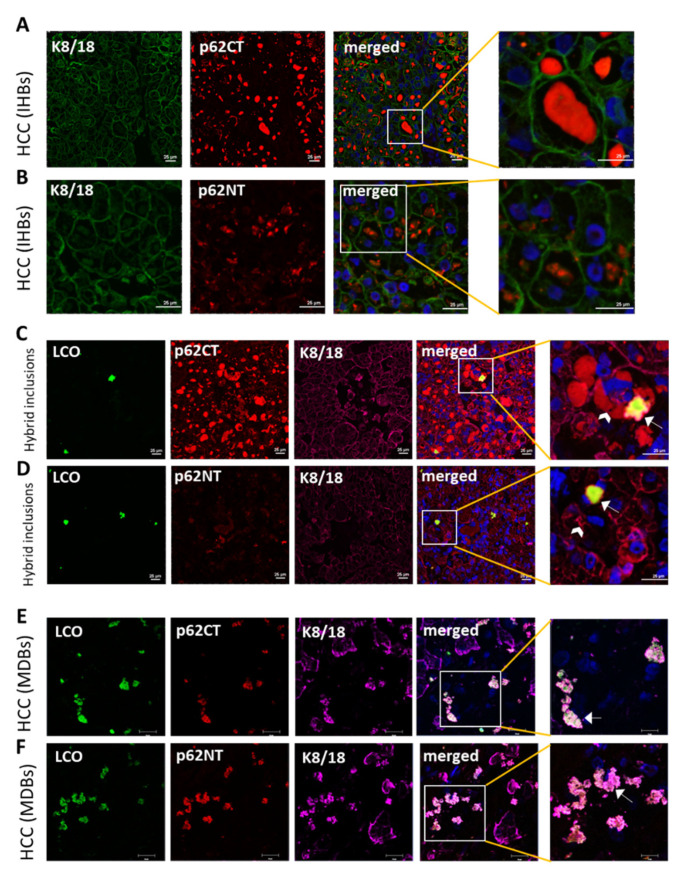
The p62-H2 contributed to IHB formation in human HCC. Frozen tissue sections from human HCC were stained for double-label immunofluorescence microscopy with antibodies against the (**A**) C-terminus (p62CT) or the (**B**) N-terminus (p62NT) of p62 (red) combined with K8/18 labeling (green) and DAPI (blue) staining. (**C**,**D**) HCC containing IHBs and hybrid inclusions were stained for triple-label immunofluorescence microscopy with antibodies against p62CT (red in **C**) or p62NT (red in **D**) and K8/18 (magenta in **C** and **D**) and the LCO dye (green) and DAPI (blue). (**C**,**D**) The arrows indicate aggregate with co-localization of p62, keratins, and an LCO fluorescence signal as characteristic of a hybrid lesion, (**C**) whereas the arrowheads denote IHBs containing only p62-H2 and (**D**) granular p62-H1. (**E**,**F**) HCCs containing MDBs were stained for triple-label immunofluorescence microscopy with antibodies against p62CT (red in **E**) or p62NT (red in **F**) and K8/18 (magenta in **E** and **F**) and the LCO dye (green) and DAPI (blue). (**E**,**F**) The arrows indicate aggregate with co-localization of p62, keratins, and an LCO fluorescence signal as characteristic of MDBs. Scale bar = 25 μm. (**G**) Analysis of p62 aggregate size in IHBs, hybrid inclusions, and MDB samples labeled for p62CT (CT) or p62NT (NT) (IHB_CT, *n* = 10; IHB_NT, *n* = 10; MDB_CT, *n* = 10; MDB_NT, *n* = 10; Hybrid_CT, *n* = 6; Hybrid_NT, *n* = 8). (**H**) Co-localization of p62CT (CT) or p62NT (NT) with K8/18 in IHB, hybrid inclusions, and MDB samples (IHB_CT, *n* = 10; IHB_NT, *n* = 10; MDB_CT, *n* = 10; MDB_NT, *n* = 10; Hybrid_CT, *n* = 6; Hybrid_NT, *n* = 8). (**I**) Triple co-localization analysis of K8/18 and p62CT/NT overlaying aggregates with LCO staining in IHB, hybrid inclusions, and MDB samples (IHB_CT, *n* = 10; IHB_NT, *n* = 10; MDB_CT, *n* = 10; MDB_NT, *n* = 10; Hybrid_CT, *n* = 6; Hybrid_NT, *n* = 8). Single-image data are shown as scatterplot with mean +/–SEM in red. Using the Kolmogorov–Smirnov test for normality, the results showed normal distribution. Significant differences were assessed by ANOVA and Bonferroni post hoc tests and presented as specific *p*-values (* *p* ≤ 0.05). (**J**) Schematic drawing of the distinctive roles of p62 isoforms involved in IHBs, hybrid inclusions, and MDB formation.

## Data Availability

The data presented in this study are available within the article or in the Appendix A.

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
