# Peer review of "Different Roles of p62 (SQSTM1) Isoforms in Keratin-Related Protein Aggregation"

_ijms, 2021, doi:10.3390/ijms22126227_

Round 1

Reviewer 1 Report

Major concern:

The authors have qualified the colocalization data for Figures 2 and 3. However, no qualification for the following colocalization experiments.

Minor concerns:

The manuscript contains many writing problems. The authors should carefully revise and proofread the manuscript throughout. Some examples:

  1. Abstract, last sentence, no definitions for K8 and K18 (I suggest to use the full name but not the abbreviations). No need to mention the abbreviation of LCO since it only appears one time in the Abstract. Moreover, the Abstract lacks a sentence to summarize the conclusion of this work.
  2. Line 65 and other places, it should be HEK 293T and HeLa cells
  3. Figures 4 and 5, it is difficult to distinguish the staining of p62 and K8/K18 shown in red and in magenta in the merge lane. It is better to use different colors, i.e., one in red and one in cyan/blue/etc.
  4. Line 197, the subtitle is misleading. The current research could not tell the nature of the interaction is beta-sheet or ubiquitin-dependent since only colocalization experiments were performed. Furthermore, ubiquitin is appeared in aggregates containing both p62-H1 and -H2 and why only H1 is mentioned.
  5. Line 242, both panels of K and L are cotransfected with Ub. Any evident for the statement that co-localizes with K18 regardless of the presence or absence of ubiquitin.
  6. Line 568 and other places, the authors used mouse-derived cell lines and human tissue samples. I think it is not in vivo experiments since the authors did not use living organisms but only derived samples. Maybe it is better to described as ex vivo. Furthermore, no statements for the approvements and approved numbers for the animal or human studies by the Institutional Review Board.

Author Response

We are grateful for the comments and the statement that we have addressed most of the reviewer’s concerns during last round of review.

The authors have qualified the colocalization data for Figures 2 and 3. However, no qualification for the following colocalization experiments.

Reply: We have performed additional morphometric quantification of experiments shown in Figures 4, 5 and 9. The experiments in PLC/PRF/5 and MEFs (Figures 6, 7 and 8) are control experiments to exclude possible interference of endogenous keratins and p62. Unfortunately, the slides of these experiments have bleached which did not allow reanalysis and quantification.

The manuscript contains many writing problems. The authors should carefully revise and proofread the manuscript throughout. Some examples:

  1. Abstract, last sentence, no definitions for K8 and K18 (I suggest to use the full name but not the abbreviations). No need to mention the abbreviation of LCO since it only appears one time in the Abstract. Moreover, the Abstract lacks a sentence to summarize the conclusion of this work.

Reply: The abstract has been revised and a conclusion added.

  1. Line 65 and other places, it should be HEK 293T and HeLa cells

Reply: Corrected

  1. Figures 4 and 5, it is difficult to distinguish the staining of p62 and K8/K18 shown in red and in magenta in the merge lane. It is better to use different colors, i.e., one in red and one in cyan/blue/etc.

Reply: We replaced magenta with blue in Figures 4 and 5 in the revised manuscript.

  1. Line 197, the subtitle is misleading. The current research could not tell the nature of the interaction is beta-sheet or ubiquitin-dependent since only colocalization experiments were performed. Furthermore, ubiquitin is appeared in aggregates containing both p62-H1 and -H2 and why only H1 is mentioned.

Reply: We thank the reviewer for raising this point. We have changed the subtitle to “β-sheet-conformation is associated with p62-H2 and K8 co-aggregates” and have included an extra subtitle concerning ubiquitin. We hope that this change now better refers to the observed findings.

  1. Line 242, both panels of K and L are cotransfected with Ub. Any evident for the statement that co-localizes with K18 regardless of the presence or absence of ubiquitin.

Reply: We clarified this issue by adding the explanation “Furthermore, K18 aggregates co-localized with p62-H1 in the situation of overexpressed ubiquitin (Figure 5 J). This co-aggregation of K18 with p62-H1 was markedly more pronounced than without overexpression of ubiquitin (compare with Figures 3 and 4).”

  1. Line 568 and other places, the authors used mouse-derived cell lines and human tissue samples. I think it is not in vivo experiments since the authors did not use living organisms but only derived samples. Maybe it is better to described as ex vivo. Furthermore, no statements for the approvements and approved numbers for the animal or human studies by the Institutional Review Board.

Reply: We thank the reviewer for this comment and have replaced in vivo by the term in human tissue. The approval of the animal experiments has already been mentioned the manuscript (see L521 to L524).

Reviewer 2 Report

The authors investigated the interaction of the two human p62 isoforms, p62-H1 and p62-H2, with keratins-8 and 18. The results seemed to be interesting. However, following points should be considered.

  1. Protein interaction should be validated with a co-IP method. Right now, only a lot of images were presented. These results might not support a solid conclusion.
  2. Figure 2M, comparisons between groups seemed to be unclear.
  3. All figures should be indicated with sample number and a statistical analysis should be conducted.
  4. Conclusion was consisted of 6 points and these points  seemed to be diffuse and cannot work together to draw a brief and focus conclusion.
  5. “***p ≤ 0.05” or“*p ≤ 0.05” should be consistent in figure legends.

Author Response

  1. Protein interaction should be validated with a co-IP method. Right now, only a lot of images were presented. These results might not support a solid conclusion.

Reply: We agree with the reviewer that further data should support the protein interactions. As p62 and keratin form aggregates, a solubilization step is needed for co-IP which might interfere with the protein interactions. We have therefore chosen FRET as an additional technology to further investigate protein interaction in co-aggregates. While FRET cannot detect direct interaction of proteins, the high local proximity of FRET donor and acceptor with a distance of 1-15 nm which is essential to detect FRET makes a direct interaction of p62 and Keratin8/18 most likely. This aspect is now included in the discussion.

  1. Figure 2M, comparisons between groups seemed to be unclear.

Reply: We have now performed the multiple comparison among the groups in the revised manuscript.

  1. All figures should be indicated with sample number and a statistical analysis should be conducted.

Reply: We have now included the sample numbers in the figure legends.

  1. Conclusion was consisted of 6 points and these points seemed to be diffuse and cannot work together to draw a brief and focus conclusion.

Reply: The conclusion has been revised and shortened.

  1. “***p ≤ 0.05” or“*p ≤ 0.05” should be consistent in figure legends.

Reply: The figure legends have been corrected.

Round 2

Reviewer 2 Report

The authors have carefully revised their manuscript. This manuscript in the present revision version can be acceptable for publication in this journal.

This manuscript is a resubmission of an earlier submission. The following is a list of the peer review reports and author responses from that submission.

Round 1

Reviewer 1 Report

In this research, the authors studied the dissimilar cellular distribution patterns of two Sequestosome-1/p62 isoforms in several cell lines. The authors showed that the shorter isoform or mutant lacking the SH2 domain aggregated more seriously in the cells and proposed that the shorter isoform may play a role in diseases.

Major concerns:

  1. This work is rather preliminary. The authors have presented dissimilar subcellular distributions of the two isoforms. However, neither mechanistic insights or biological/medical implications are designed and performed. For example, what is the molecular mechanism for the observation the shorter form formed larger aggregates? Is it originated from the misfolding of p62, alterations in self-assembly or protein-protein interaction network? What is the cellular consequence or biological roles of the two isoforms underlying the different distribution? It is well-known that p62 acts as a scaffold protein delivering ubiquitinated proteins between protein aggregates, proteosomes and autophagosomes. Do the two isoforms have any difference in these cellular events? I strongly suggest the authors to extend their observations to biological/medical implications.
  2. The authors claimed that the two isoforms might play distinct roles in protein aggregation diseases without any evidence. It is suggested to check clinical samples or animal models whether the two isoforms do perform differentially in normal and pathological tissues/cells.
  3. The confocal images can only provide selected regions of limited cells. The authors should qualify the size, number or percentage of aggregate(s) as well as co-localization coefficient in a larger number of cells with repetitions.
  4. Overexpression generally produce thousands fold of exogenous proteins than the endogenous proteins. It remains unclear whether the observations of overexpressed proteins do reflect the dissimilar actions of the endogenous proteins in corresponding cellular events.

Reviewer 2 Report

The authors found that p62-H2 shows more pronounced co-localization with aggregated keratin than p62-H1 and indicated that different p62 isoforms has to be considered in future studies on p62 in protein homeostasis and protein aggregation diseases. The manuscript is interesting and well prepared. However, following points should be considered.

  1. p62-H2 could co-localize with keratin. However, no mutant site was involved. The authors need to identify possible bind site in an amino acid level between p62-H2 and keratin.
  2. Most of experiments come from images. A co-Immunoprecipitation experiment should be conducted.
  3. Sample number should be added in the figures.